inorganic chemistry

VO$_2$, hydrothermal, stripped vanadium solution, eco-friendly, high conversion rate

**Author for correspondence:**
Yimin Zhang
e-mail: zym126135@126.com

# Eco-friendly synthesis of VO$_2$ with stripped pentavalent vanadium solution extracted from vanadium-bearing shale by hydrothermal process in high conversion rate

Qian Kang[1], Yimin Zhang[1,2,3,4], Shenxu Bao[1,2] and Guobin Zhang[1]

[1]School of Resources and Environmental Engineering, and [2]Hubei Key Laboratory of Mineral Resources Processing and Environment, Wuhan University of Technology, Wuhan 430070, People's Republic of China
[3]State Environmental Protection Key Laboratory of Mineral Metallurgical Resources Utilization and Pollution Control, and [4]Hubei Collaborative Innovation Center for High Efficient Utilization of Vanadium Resources, Wuhan University of Science and Technology, Wuhan 430081, Hubei Province, People's Republic of China

QK, 0000-0002-9624-5111; SB, 0000-0003-1295-3389

VO$_2$(B) has shown excellent cathode performance in lithium batteries and become a hot research topic in recent years. A stripped vanadium solution extracted from vanadium-bearing shale containing a high concentration of vanadium and certain amounts of impurities was used as a vanadium source to synthesize VO$_2$(B) by hydrothermal process. The VO$_2$ conversion rate can reach as high as 99.47% in a reaction time of 8 h, and this is the highest result reported. The crystalline structure and morphology of the synthesized products were characterized by X-ray diffraction (XRD), Raman spectroscopy, X-ray photoelectron spectroscopy (XPS), Scanning electron microscopy (SEM) and Transmission electron microscopy (TEM). Furthermore, the electrochemical properties of VO$_2$(B) in lithium-ion batteries were investigated. The results indicated that the VO$_2$(B) has the initial specific discharge capacity of 192.0 mAh g$^{-1}$. Stripped vanadium solution is a raw material for producing V$_2$O$_5$ and NH$_4$VO$_3$, which are indispensable vanadium sources in VO$_2$ synthesis. Therefore, synthesis of VO$_2$ via hydrothermal reduction by oxalic acid using stripped vanadium solution extracted from vanadium-bearing shale as a direct vanadium

source is an eco-friendly, innovative and efficient method, and will have a great impact on VO$_2$ synthesis.

## 1. Introduction

Nowadays, the depletion of fossil fuel resources and the efforts to control environmental pollution have led to the emergence and development of new clean energy, such as solar cells, fuel cells, batteries [1–3] and supercapacitors [4–6]. Vanadium is an important transition metal and demonstrates excellent optical and electrical properties [7,8]. VO$_2$(B) in the form of a metastable monoclinic structure is an excellent cathode material for lithium-ion batteries, because of its great electrode potential and tunnel structure [9,10]. VO$_2$(B) with different morphologies of nanobelts, nanoflowers and nanoflakes was synthesized and used as an electrode material in lithium-ion batteries [11,12]. At the current density of 20 mA g$^{-1}$, the initial specific discharge capacities of the flower-like, belt-like and flake-like VO$_2$(B) were 254.0, 205.2 and 56.0 mAh g$^{-1}$, respectively [13]. VO$_2$ hollow microspheres with empty spherical cores and radially protruding nanowires were synthesized by ion modulating method. At the current density of 1 A g$^{-1}$, the initial specific discharge capacity of hollow microspheres was 163 mAh g$^{-1}$, and 73% of the initial capacity was retained after 1000 cycles, exhibiting good battery performance [14]. Novel tip-ended VO$_2$(B) nanorods were synthesized with V$_2$O$_5$ as a vanadium resource and formaldehyde as a reductant and its specific discharge capacity reached as high as 320 mAh g$^{-1}$ [15].

So far, many methods have been developed for VO$_2$ synthesis, such as chemical precipitation, solid phase reduction reaction, thermal decomposition, sol-gel, hydrothermal and laser induced vapour deposition [16]. However, the hydrothermal method with its low environmental impact is usually used to synthesize metastable VO$_2$(B). In the hydrothermal process, NH$_4$VO$_3$ and V$_2$O$_5$ are the most common vanadium sources reported. Oxalic acid [3,17,18], N$_2$H$_4$·H$_2$O [19], ethylene glycol [20,21], octadecylamine [22], citric acid [23], formic acid [24,25], and so on were usually used as reductants. Owing to its hypotoxicity and convenience, oxalic acid has become the most popular reductant in hydrothermal reactions for VO$_2$ synthesis. NH$_4$VO$_3$ and oxalic acid were used to synthesize different kinds of VO$_2$(B) nanostructures at different reaction temperatures and times via hydrothermal method [26]. VO$_2$(B) nanostructures were synthesized via hydrothermal method with V$_2$O$_5$ as a vanadium source and oxalic acid as a reductant. It was found that the obtained products contained three nanostructures of carambolas, rods and bundles, and the concentration of oxalic acid could affect their proportions [27]. However, the long reaction time, up to 24 h or even several days, is an ineffective method as it consumes too much time and energy. Besides, almost no research about the VO$_2$ conversion rate has been reported.

Stripped vanadium solution extracted from vanadium-bearing shale is an intermediate product obtained from vanadium-bearing shale through a process including roasting, leaching, purification and enrichment [28,29]. Stripped vanadium solution is usually used to produce NH$_4$VO$_3$ via vanadium precipitation by adding ammonium salt and to produce V$_2$O$_5$ via vanadium precipitation by adding ammonium salt and calcination in the vanadium-bearing shale industry [30]. Stripped vanadium solution extracted from vanadium-bearing shale containing a high concentration of vanadium, which could reach as high as 40 g l$^{-1}$ with a lower amount of impurities, is a promising vanadium source in VO$_2$ synthesis.

Compared with the previous studies, synthesis of VO$_2$ with the stripped pentavalent vanadium solution shows many advantages. It is more eco-friendly and energy efficient as it omits the preparation of V$_2$O$_5$ or NH$_4$VO$_3$ which will generate huge amounts of pollutants and great energy expenditure. The short reaction time makes this synthesis method more energy efficient. The high conversion rate of vanadium shows the method is an efficient one.

In this work, stripped vanadium solution extracted from vanadium-bearing shale was used as a new vanadium source directly and oxalic acid was used as reductant in the hydrothermal process to synthesize VO$_2$(B). The structural, morphology, purity and chemical properties of the synthesized VO$_2$ were investigated and the reason for the high conversion rate of vanadium was also explored.

## 2. Experimental procedure

### 2.1. Preparation of stripped vanadium solution

The stripped vanadium solution was obtained from the vanadium-bearing shale via a traditional process [31]. Firstly, the vanadium-bearing shale was roasted at 800°C for 1 h in muffle. Next, the roasted

**Table 1.** Main chemical composition of the stripped vanadium solution.

| items | V | Fe | Al | Mg | Na | P | Si |
|---|---|---|---|---|---|---|---|
| concentration (g L$^{-1}$) | 24.58 | 0.119 | 0.131 | 0.025 | 22.14 | 0.085 | 0.022 |

vanadium-bearing shale ore was leached under conditions of $H_2SO_4$ (v/v) at 15%, temperature of 95°C, liquid/solid ratio of 1.5 and time of 6 h, and a yellow leaching solution was produced after filtration. Then, moderate oxidant $NaClO_3$ was added into the leaching solution to oxidize vanadium to V(+5), and NaOH solution of 40 g l$^{-1}$ was used to adjust the pH to about 1.8. Afterwards, a three-stage extraction process was conducted to achieve the loaded organic phase under the following conditions: organic solution consisting of 20% trialkylamine (N235)(v/v) and 5% tri-butyl phosphate (v/v) mixed with 75% acidized kerosene (v/v), A/O phase ratio of 3 and equilibrium time of 5 min. After that, the loaded organic phase was scrubbed with equal volume $H_2SO_4$ (v/v) of 3% for three stages, followed by deionized water to achieve the organic phase pH > 4. Finally, a colourless stripped vanadium solution was obtained after the scrubbed loaded organic phase underwent a four-stage strip with the following conditions: stripping agent of NaOH solution of 40 g l$^{-1}$, O/A radio of 6 and equilibrium time of 6 min. The main chemical composition of the stripped vanadium solution is listed in table 1.

## 2.2. Synthesis of VO$_2$(B)

In a typical synthesis, 60 ml of stripped vanadium solution was measured and its pH was adjusted to 2 by adding sulfuric acid dropwise with continued magnetic stirring. Next, 3.65 g of $H_2C_2O_4 \cdot 2H_2O$ was dissolved into the above solution, and a yellow-green solution was obtained. Then, the solution was transferred into a Teflon-lined stainless steel autoclave. The autoclave was sealed and placed in an oven for hydrothermal treatment at 220°C for 8 h. After the autoclave was cooled down to room temperature, the precipitate was filtrated and washed with deionized water and ethanol. Then, the blue-black powders were dried at 80°C for 6 h.

## 2.3. Data treatment

The yield of VO$_2$ (R) was calculated based on equation (2.1). In the equation $C_R$ and $C_F$ represent the concentrations of vanadium in residue liquid and feed solution, respectively. $V_R$ and $V_F$ represent the volume of residue liquid and feed solution, respectively.

$$R = \frac{1 - C_R V_R}{C_F V_F} \times 100. \tag{2.1}$$

## 2.4. Characterization

The chemical compositions of the stripped solution and synthesized VO$_2$ were analysed by inductively coupled plasma−optical emission spectroscopy (ICP−OES, Optima 4300DV, Perkin−Elmer, USA). The crystal structure of the product was studied by a D/MAX-RB X-ray diffractometer (XRD, Rigaku, Japan), with Cu K$\alpha$ radiation ($\lambda$ = 0.15406 nm) in the $2\theta$ range of 10−70°. The morphology of synthesized powder was characterized by field emission scanning electron microscopy (SEM, JEOL, Japan) performed at 5.0 kV. The local structure of the powder was analysed by Raman scattering (RENISHAW Raman microscope, InVia, UK) with an argonion laser as the excitation light source at $\lambda$ = 633 nm. X-ray photoelectron spectroscopy (XPS) measurement was recorded by a VG ESCALAB 210 electron spectrometer (UK). Transmission electron microscopy (TEM) was conducted by JEOL JEM-2100F (Japan). The Brunauer−Emmett−Teller (BET) surface area was determined by Micromeritics ASAP 2020 nitrogen adsorption apparatus (USA).

## 2.5. Electrochemical measurements

The electrochemical performance of the obtained VO$_2$(B) was characterized by coin cell assembly (2016-type). The cathode electrode was made up of a ground mixture of 70% active material, 20% acetylene black and 10% polyvinylidene fluoride (10 wt%) dissolved in N-methylpyrrolidone solution. After an

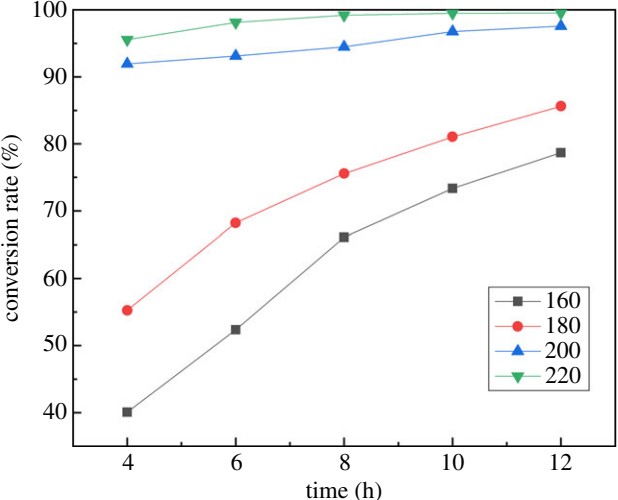

**Figure 1.** Effects of time and temperature on conversion rate.

ultrasonic oscillation of 30 min, the mixture was coated uniformly on 10-μm thick aluminium foil and dried overnight at 90°C to yield the working electrodes. Then, the aluminium foil was punched into circular disc with a diameter of 8 mm. The mass loading of the active material was 1.5–2 mg cm$^{-2}$. In a typical test, coin cell (2025) was assembled in an argon-filled glove box. Celgard 2325 was used as a separator membrane. LiPF6 of 1 mol l$^{-1}$ dissolved in ethylene carbonate, dimethyl carbonate and ethylene methyl carbonate in 1 : 1 : 1 (volumetric ratio) was used as the electrolyte solution. Lithium metal foil was used as the counter electrode. The cells were aged for 12 h before charge/discharge to ensure full absorption of the electrolyte into the electrodes. Galvanostatic charge/discharge measurement was performed by a multichannel battery testing system (LAND CT2001A) at a current density of 100 mAg$^{-1}$ with a voltage region between 0.01 and 3.0 V. The cyclic voltammetry (CV) and electrochemical impedance spectroscopy were tested on an Autolab Potentiostat Galvanostat with a frequency ranging from 100 kHz to 0.01 Hz. All the measurements were conducted at room temperature.

# 3. Results and discussion

## 3.1. VO$_2$ hydrothermal synthesis

### 3.1.1. Effect of time and temperature on VO$_2$ conversion rate

In the hydrothermal synthesis process, temperature and time will greatly affect the product conversion rate, so the influences of time ranging from 4 h to 12 h and temperature from 160°C to 220°C on the product conversion rate were investigated. As shown in figure 1, both the time and temperature have a significant positive effect on conversion rate. The conversion rate rises rapidly as the temperature increases. The hydrothermal reduction reaction is endothermic, therefore, an increase of temperature improves the progress of the reaction. The conversion rate continues to rise as the process time increases. The synthesis of VO$_2$ is a staged process in which V(V) is first reduced to V(?) and then grows into VO$_2$ crystals. Prolonging time is beneficial for the reduction and formation of VO$_2$ crystals. Conversion rates at temperatures of 200°C and 220°C are much higher than those at temperatures of 160°C and 180°C. This is because the decomposition temperature of oxalic acid is 189.5°C. When the temperature is higher than this, oxalic acid decomposes rapidly and the reduction speed is faster. Therefore, the conversion rate is higher.

### 3.1.2. Effect of vanadium concentration on VO$_2$ conversion rate

In a typical hydrothermal synthesis of VO$_2$, the Cv (vanadium concentration) is between 2.5 and 11.2 g l$^{-1}$, and the reaction time can range from 24 h to several days. Compared with the synthesis methods in other papers, the Cv in our studies is as high as 24.58 g l$^{-1}$, so the effect of Cv on the VO$_2$ conversion rate was studied under a reaction temperature of 493 K, reaction time of 8 h, and

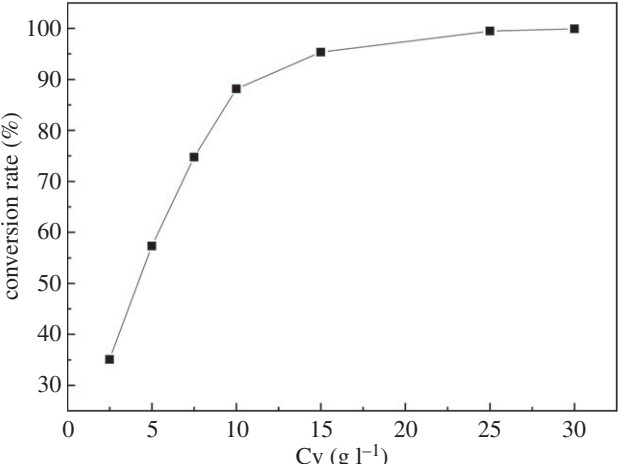

**Figure 2.** Effects of Cv on conversion rate.

**Table 2.** Chemical composition of as-synthesized VO$_2$.

| items | VO$_2$ | Fe | Al | P | Na$_2$O | S | Si |
|---|---|---|---|---|---|---|---|
| content (wt%) | 99.52 | 0.09 | 0.11 | 0.08 | 0.12 | 0.02 | 0.06 |

molar ratio of oxalic acid to vanadium of 1. As figure 2 shows, the VO$_2$ conversion rate increased with the increase of Cv and it increased slightly when the Cv was higher than 25 g l$^{-1}$. Vanadium yield can reach up to 99.47% when the Cv is 25 g l$^{-1}$. Based on the above result, the reason why the VO$_2$ conversion rate is so high in a much shorter reaction time is the high vanadium concentration in stripped pentavalent vanadium solution extracted from vanadium-bearing shale. In the vanadium-bearing shale industry, the vanadium concentration of stripped vanadium solution can easily reach 25 g l$^{-1}$, so synthesis of VO$_2$ by this method is efficient.

## 3.2. Purity characterization

As the main chemical composition of the stripped vanadium solution listed in table 1, there are some impurities in the vanadium source. Therefore, the purity of the synthesized VO$_2$ must be studied. The chemical composition of the synthesized VO$_2$ is shown in table 2. It can be seen that the purity of the VO$_2$ synthesized is 99.52%, and little impurities also exist in the product.

## 3.3. Structure and morphology

Figure 3 shows the XRD pattern of the synthesized VO$_2$(B). Diffraction peaks in the product at 14.3°, 15.4°, 17.6°, 25.3°, 28.6°, 29.0°, 30.2°, 33.9°, 38.1°,44.1°, 45.1°, 49.5°, 53.9° and 59.4° are in accord with those of (001), (200), (−201), (110), (−202), (002), (−401), (−311), (401), (003), (−511), (312), (601) and (−404) crystal planes in VO$_2$(B) (JCPDS card No. 65-7960) with lattice constants of $a = 12.03$ Å, $b = 3.693$ Å, $c = 6.42$ Å and $\beta = 106.6°$. No peaks of other substances were found, indicating that only VO$_2$(B) crystal exists. The strong and sharp diffraction peaks demonstrate that the VO$_2$(B) is well crystallized and preferentially orientated along (110). The results of XRD suggest that the synthesis condition including precursor dosage, pH, temperature and time is suitable for VO$_2$(B) synthesis.

Figure 4 demonstrates the Raman spectrum of the synthesized VO$_2$(B). The Raman spectrum characterized by the peaks situated at 192, 282, 405, 498, 693 and 991 cm$^{-1}$ belong to the Raman signature of VO$_2$ (B). The band located at 282 cm$^{-1}$ can be attributed to the V$_3$−O bridging flexural vibration. The band located at 405 cm$^{-1}$ can be classified as the V−O flexural vibration. A broad band in the range of 400–600 cm$^{-1}$ is due to V$_2$−O bridging flexural vibration. The bands at 693 cm$^{-1}$ and 991 cm$^{-1}$ are assignable to V−O and V=O stretching vibration, respectively [32–34].

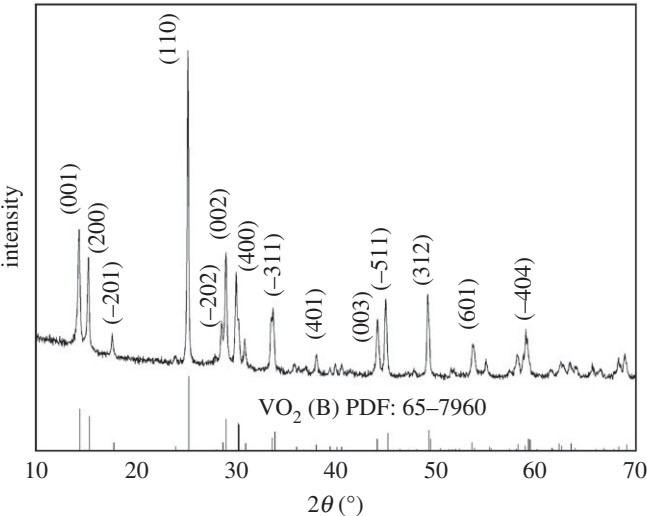

**Figure 3.** XRD of as-synthesized VO₂.

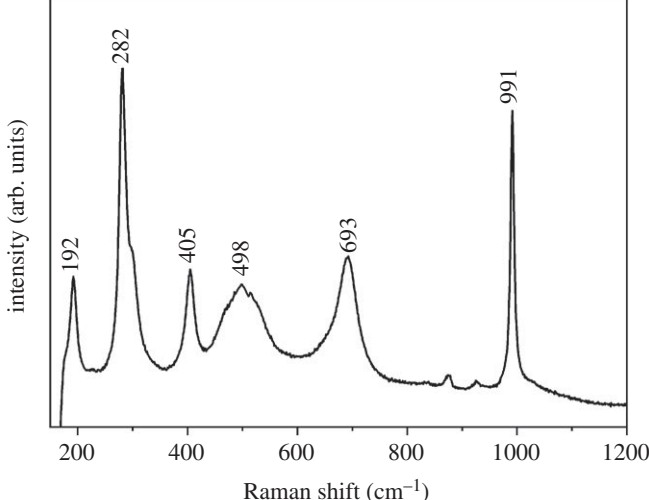

**Figure 4.** Raman spectrum of as-synthesized VO₂.

To further measure the composition of the product, XPS analysis was performed and the results shown in figure 5. The wide-survey XPS spectrum (figure 5*a*) demonstrates that the peaks can be assigned to C, N, V and O. The V 2p3/2 peak and the O1 s peak of the sample (figure 5*b*) present the vanadium valence of $V^{4+}$ (516.6 eV) and $O^{2-}$ (529.4 eV) [22,35]. According to the relationship between the atomic number and the element peak area and atomic mass, we get the ratio of the atomic number of O to V close to 2, thus indicating that the synthesized product is $VO_2$.

Figure 6 displays SEM images of the synthesized $VO_2(B)$. The $VO_2(B)$ exist in some irregular aggregates which are composed of numerous randomly self-assembled nanorods. The average diameter of the aggregates is about 15 µm. The width and length of the rods ranges from 80 to −160 nm and from 1 to 2 µm, respectively.

As shown in figure 7, the TEM images (figure 7*a*,*b*) of $VO_2(B)$ exhibit a nanorods structure. The HRTEM image (figure 7*d*) shows that the lattice fringes correspond to crystalline $VO_2(B)$. The interplanar distances of 0.29 nm, 0.35 nm and 0.37 nm agree well with the (400), (110) and (201) crystal planes of $VO_2(B)$, respectively. The corresponding SAED pattern (figure 7*c*) exhibits good crystallinity.

## 3.4. Reaction mechanism

The existing form of V(V) in a solution whose pH is about 2 is $H_2V_{10}O_{28}^{4-}$ [36]. In a hydrothermal experiment, oxalic acid is a reductant for converting $H_2V_{10}O_{28}^{4-}$ into $VO_2(B)$. In the initial stage of the

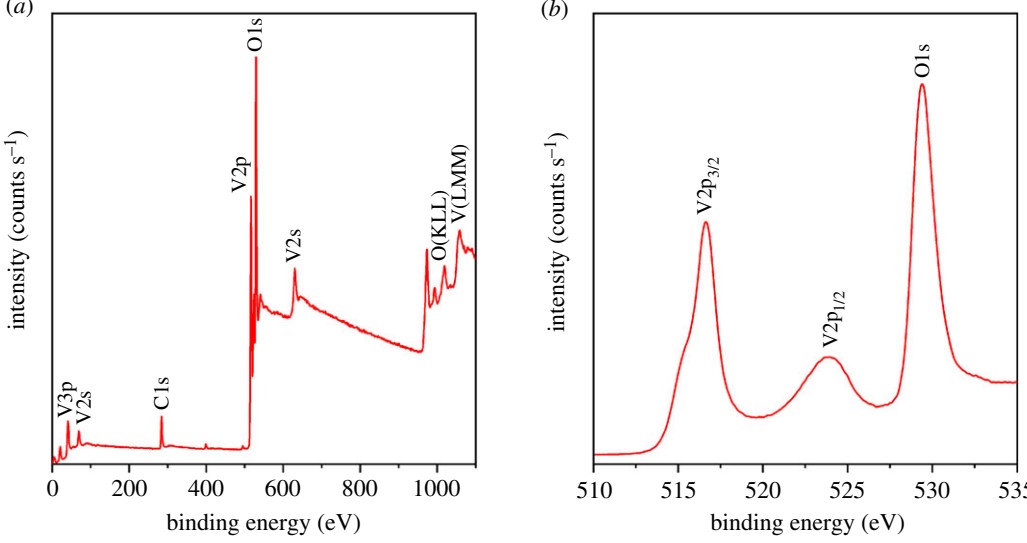

**Figure 5.** XPS spectrum of as-synthesized VO$_2$: (a) survey spectrum and high-resolution, (b) V2p and O1s.

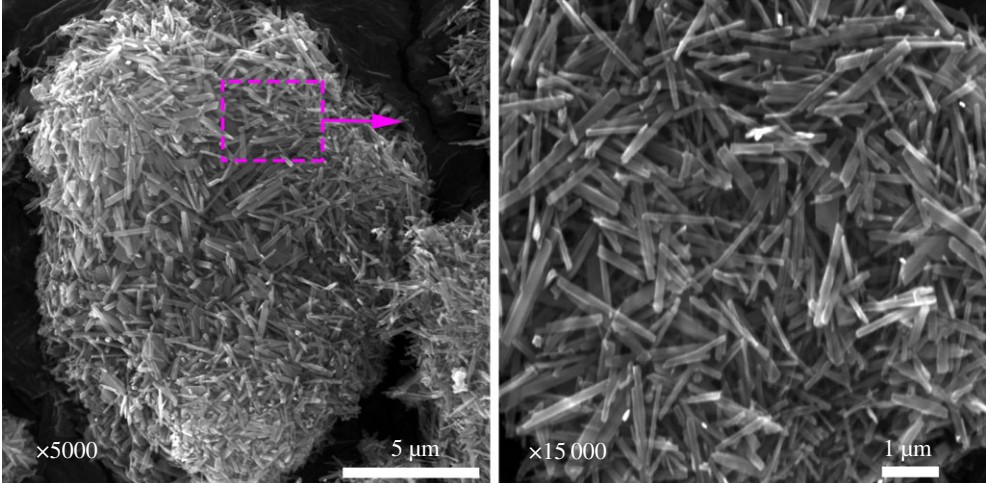

**Figure 6.** SEM images of as-synthesized VO$_2$ (a ×5000, b ×15 000).

reaction, the V$^{+5}$ in H$_2$V$_{10}$O$_{28}^{4-}$ was reduced to V$^{+4}$, then, with the temperature and pressure (self-generated pressure resulting from temperature rise) increased, V$^{4+}$ crystallized into VO$_2$. The probable reaction happened during the hydrothermal process of stripped vanadium solution extracted from vanadium-bearing shale with oxalic acid was inferred to be the following equation:

$$H_2V_{10}O_{28}^{4-} + 10H_2C_2O_4 + 4H^+ \rightarrow 10VO_2 + 15CO_2 + 5CO + 13H_2O. \tag{4.1}$$

## 3.5. Electrochemical performance

As shown in figure 8, the electrochemical performance of VO$_2$(B) nanorods is characterized. The CV curve (figure 8a) of VO$_2$(B) nanorods characterizes the phase transformation and ionic diffusion process during the reaction. VO$_2$(B) nanorods exhibit only one pair of well-defined redox peaks appearing on both electrodes (1.31/1.83 V), corresponding to the reversible transformation of V$^{3+}$/V$^{4+}$ [37]. The initial discharge capacity was 192.3 mAh g$^{-1}$ and the specific discharge capacity reduced to 95 mAh g$^{-1}$ after 20 cycles (figure 8b). After 1000 cycles, the specific capacity of nanorods reduced to only 50 mAh g$^{-1}$ (figure 8c). The Coulombic efficiency at different currents is no less than 94% from the second cycle to the 40th, and the enhanced Li insertion properties were relevant to the increased crystallinity of the material (figure 8d). The impedance plot (figure 8e) shows one semicircle in the medium-frequency region which could be due to the charge transfer resistance and a line in the

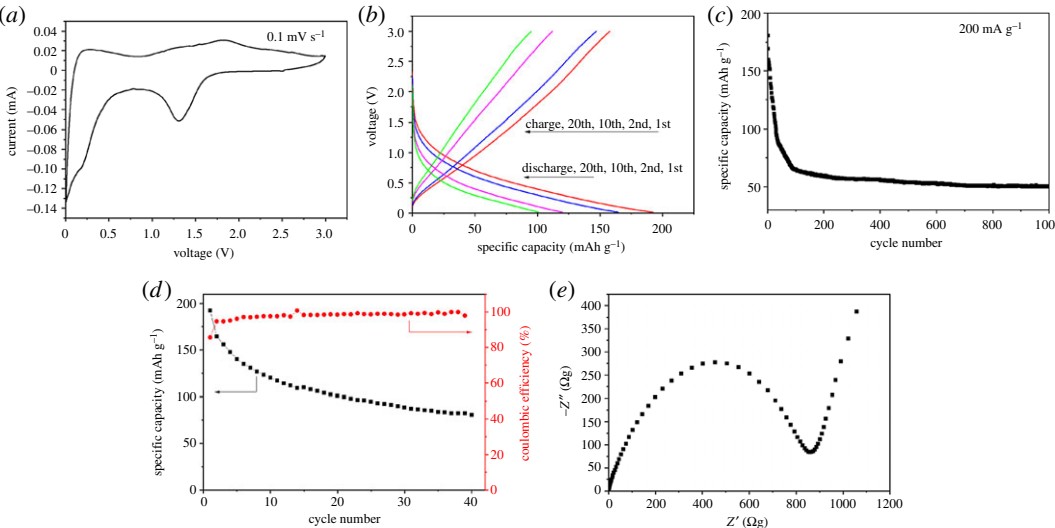

**Figure 7.** (*a,b*) TEM images, (*c*) SAED pattern, (*d*) HRTEM image of as-synthesized VO₂.

**Figure 8.** Electrochemical characterizations of as-synthesized VO₂(B) nanorods. (*a*) CV curve in 0.01 − 3 V at a scan rate of 0.1 mV s⁻¹. (*b*) Charge–discharge curves at current density of 100 mA g⁻¹ in 0.01 – 3.0 V. (*c*) Long cycling performance at current density of 200 mA g⁻¹ in 0.01 – 3 V. (*d*) Cycling performance at 100 mA g⁻¹ in 0.01 − 3 V. (*e*) AC impedance plots.

low-frequency range which could be attributed to Warburg impedance. The charges transfer resistance (Rct) of $VO_2(B)$ nanorods is about $900\,\Omega$, suggesting their slow electronic mobility [13]. The low capacity retention of the $VO_2(B)$ nanorods electrode is related to the aggregation and stress-induced structural collapse when Li-ion is inserted into and deinserted from the electrode [38]. The specific discharge capacity, which indicates the high utilization rate of cathode material, is dependent on the adsorption area to electrolyte and lithium-ion diffusion and electron transportation distance [39]. The Brunauer–Emmerr–Teller (BET) surface area of $VO_2(B)$ nanorods is $19.8\,m^2\,g^{-1}$. There is enough contact surface area for electrode–electrolyte contact. However, large specific surface areas and high surface energy also cause $VO_2(B)$ nanorods materials to aggregate together easily during charge–discharge testing, thereby increasing the charge transfer resistance and resulting in poor cycling performance [13,18].

## 4. Conclusion

In summary, $VO_2(B)$ nanorods were synthesized by an efficient and eco-friendly hydrothermal process with stripped vanadium solution extracted from vanadium-bearing shale as a vanadium source. The effects of reaction time and temperature on the conversion rate of vanadium in synthesis was first studied. The high vanadium concentration in stripped vanadium solution extracted contributes to the high $VO_2$ conversion based on the effect of Cv on conversion study. A possible reaction mechanism for the formation of $VO_2(B)$ has been put forward. The XRD, Raman spectrum and XPS characterization show the as-synthesized $VO_2$ is a pure $VO_2$ phase. The SEM and TEM results illustrate the as-synthesized $VO_2$ is present as aggregates consisting of nanorods. As cathode electrodes in Li-ion batteries, nanorods of $VO_2(B)$ exhibited an initial specific capacity of 192.3 $mAh\,g^{-1}$ and their capacity decreased to 95 $mAh\,g^{-1}$ after 20 cycles. After 1000 cycles, only 50 $mAh\,g^{-1}$ was retained. The poor cycling performance of the material was due to the aggregation in the test. Our study illustrates $VO_2(B)$ nanorods synthesized by this method can be a promising cathode electrode material in Li-ion batteries, but more research is needed to improve the electrochemical performance of the material. Stripped vanadium solution extracted used as a new vanadium source in $VO_2$ synthesis can omit the process of its conversion into $NH_4VO_3$ or $V_2O_5$, thereby reducing environmental pollution and energy consumption.

Data accessibility. This article does not contain any additional data.

Authors' contributions. Q.K., S.X.B. and Y.M.Z. conceived and design; Q.K. performed the experiments and drafted the article. S.X.B., Y.M.Z. and G.B.Z. coordinated the study and revised the article. All authors gave final approval for publication.

Competing interests. There have no competing interest.

Funding. his research was supported by National Natural Science Foundation of China (51404177) and National Key Science-Technology Support Programs of China (2015BAB03B05).

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
