## [Reviewer comments · Royal Society Open Science]

Review History

RSOS-181116.R0 (Original submission)

Review form: Reviewer 1

Is the manuscript scientifically sound in its present form?

No

Are the interpretations and conclusions justified by the results?

No

Is the language acceptable?

Yes

Is it clear how to access all supporting data?

Yes

Do you have any ethical concerns with this paper?

No

Have you any concerns about statistical analyses in this paper?

No

Recommendation?

Major revision is needed (please make suggestions in comments)

Comments to the Author(s)

Review attached (Appendix A).

Review form: Reviewer 2 (Sang-Jae Kim)

Is the manuscript scientifically sound in its present form?

No

Are the interpretations and conclusions justified by the results?

Yes

Is the language acceptable?

No

Is it clear how to access all supporting data?

Not Applicable

Do you have any ethical concerns with this paper?

No

Have you any concerns about statistical analyses in this paper?

No

Recommendation?

Major revision is needed (please make suggestions in comments)

Comments to the Author(s)

Eco-friendly synthesis of VO₂ with stripped pentavalent vanadium solution extracted from vanadium-bearing shale by hydrothermal in high yield

Manuscript ID: RSOS-181116

After careful reading this manuscript, I cannot recommend this to be published in Royal society open science at the current form. However, it can be published after major revision if the authors addressed all the issues raised.

Major Revision:

1. The experimental methodologies and material preparation are not new. Hence, a clear statement to the novelty of the work presented is necessary. Further, how did you choose 1073K for 1 h (stripped vanadium) and 220° C for 8h (VO₂)? Are it's optimized conditions?
2. The quality of fig. 3 (SEM pictures) is not sufficient for the publication
3. The electrode materials are characterized using a half-cell test configuration which is quite routine. Hence, authors need to provide the full device characterization for making the

manuscript publishable.

4. The author had prepared their electrode material using 20 % acetylene black which will have a massive impact on the electrochemical measurement, So, we can't able to justify the true nature of the VO₂(B) as the cathode material.

5. The author needs to elaborate on the electrochemical method in the revised manuscript.

6. The mass loading of the electrode needs to be provided in the revised manuscript.

7. Surface area and pore size analysis plays a vital role in the electrochemical properties of any material. Authors need to provide the BET analysis of the VO₂ in the revised manuscript.

8. Long-term cyclic stability is one of the main aspects of electrode material to implement in the practical applications. so author needs to perform cyclic stability in the revised manuscript.

9. The charge transfer kinetics and to evaluate the capacitive nature of the VO₂ electrode material EIS analysis need to be performed in the revised manuscript.

10. Some literature about novel energy storage devices should be cited in suitable places.

References:

1) Journal of Industrial and Engineering Chemistry 64, (2018), 134-142.

2) Current Applied Physics 9(6), (2009), 1195-1198.

3) ChemElectroChem 4(12), (2017), 3302-3308.

4) Electrochimica Acta 56(5), (2011), 2122-2126.

5) Cryst. Res. Technol. 46, (2011), 507 - 510.

Decision letter (RSOS-181116.R0)

13-Nov-2018

Dear Dr Kang:

Title: Eco-friendly synthesis of VO₂ with stripped pentavalent vanadium solution extracted from vanadium-bearing shale by hydrothermal in high yield

Manuscript ID: RSOS-181116

The editor assigned to your manuscript has now received comments from reviewers. We would like you to revise your paper in accordance with the referee and Subject Editor suggestions which can be found below (not including confidential reports to the Editor). Please note this decision does not guarantee eventual acceptance.

Please submit your revised paper before 06-Dec-2018. Please note that the revision deadline will expire at 00.00am on this date. If we do not hear from you within this time then it will be assumed that the paper has been withdrawn. In exceptional circumstances, extensions may be possible if agreed with the Editorial Office in advance. We do not allow multiple rounds of revision so we urge you to make every effort to fully address all of the comments at this stage. If deemed necessary by the Editors, your manuscript will be sent back to one or more of the original reviewers for assessment. If the original reviewers are not available we may invite new reviewers.

To revise your manuscript, log into <http://mc.manuscriptcentral.com/rsos> and enter your Author Centre, where you will find your manuscript title listed under "Manuscripts with Decisions." Under "Actions," click on "Create a Revision." Your manuscript number has been

appended to denote a revision. Revise your manuscript and upload a new version through your Author Centre.

Please also include the following statements alongside the other end statements. As we cannot publish your manuscript without these end statements included, if you feel that a given heading is not relevant to your paper, please nevertheless include the heading and explicitly state that it is not relevant to your work.

- Ethics statement

Please clarify whether you received ethical approval from a local ethics committee to carry out your study. If so please include details of this, including the name of the committee that gave consent in a Research Ethics section after your main text. Please also clarify whether you received informed consent for the participants to participate in the study and state this in your Research Ethics section.

OR

Please clarify whether you obtained the necessary licences and approvals from your institutional animal ethics committee before conducting your research. Please provide details of these licences and approvals in an Animal Ethics section after your main text.

OR

Please clarify whether you obtained the appropriate permissions and licences to conduct the fieldwork detailed in your study. Please provide details of these in your methods section.

- Acknowledgements

On behalf of the Subject Editor Professor Anthony Stace and the Associate Editor Professor Eva Hevia.

RSC Associate Editor:
Comments to the Author:

(There are no comments.)

RSC Subject Editor:

Comments to the Author:

(There are no comments.)

Reviewers' Comments to Author:

Reviewer: 1

Comments to the Author(s)

Review attached.

Reviewer: 2

Comments to the Author(s)

Eco-friendly synthesis of VO₂ with stripped pentavalent vanadium solution extracted from vanadium-bearing shale by hydrothermal in high yield

Manuscript ID: RSOS-181116

After careful reading this manuscript, I cannot recommend this to be published in Royal society open science at the current form. However, it can be published after major revision if the authors addressed all the issues raised.

Major Revision:

1. The experimental methodologies and material preparation are not new. Hence, a clear statement to the novelty of the work presented is necessary. Further, how did you choose 1073K for 1 h (stripped vanadium) and 220° C for 8h (VO₂)? Are it's optimized conditions?
2. The quality of fig. 3 (SEM pictures) is not sufficient for the publication
3. The electrode materials are characterized using a half-cell test configuration which is quite routine. Hence, authors need to provide the full device characterization for making the manuscript publishable.
4. The author had prepared their electrode material using 20 % acetylene black which will have a massive impact on the electrochemical measurement, So, we can't able to justify the true nature of the VO₂(B) as the cathode material.
5. The author needs to elaborate on the electrochemical method in the revised manuscript.
6. The mass loading of the electrode needs to be provided in the revised manuscript.
7. Surface area and pore size analysis plays a vital role in the electrochemical properties of any material. Authors need to provide the BET analysis of the VO₂ in the revised manuscript.
8. Long-term cyclic stability is one of the main aspects of electrode material to implement in the practical applications.so author needs to perform cyclic stability in the revised manuscript.
9. The charge transfer kinetics and to evaluate the capacitive nature of the VO₂ electrode material EIS analysis need to be performed in the revised manuscript.
10. Some literature about novel energy storage devices should be cited in suitable places.

References:

- 1) Journal of Industrial and Engineering Chemistry 64, (2018), 134-142.
- 2) Current Applied Physics 9(6), (2009), 1195-1198.
- 3) ChemElectroChem 4(12), (2017),3302-3308.
- 4) Electrochimica Acta 56(5), (2011), 2122-2126.
- 5) Cryst. Res. Technol. 46, (2011), 507 - 510.

Author's Response to Decision Letter for (RSOS-181116.R0)

See Appendix B.

RSOS-181116.R1 (Revision)

Review form: Reviewer 1

Is the manuscript scientifically sound in its present form?

Yes

Are the interpretations and conclusions justified by the results?

Yes

Is the language acceptable?

Yes

Is it clear how to access all supporting data?

Not Applicable

Do you have any ethical concerns with this paper?

No

Have you any concerns about statistical analyses in this paper?

No

Recommendation?

Accept as is

Comments to the Author(s)

Authors have addressed all the comments and now paper is suitable to accept as is.

Review form: Reviewer 2 (Sang-Jae Kim)

Is the manuscript scientifically sound in its present form?

Yes

Are the interpretations and conclusions justified by the results?

Yes

Is the language acceptable?

Yes

Is it clear how to access all supporting data?

Yes

Do you have any ethical concerns with this paper?

No

Have you any concerns about statistical analyses in this paper?

No

Recommendation?

Accept as is

Comments to the Author(s)

Authors have answered all my question in the peer review time. Hence, I recommend it for publication.

Decision letter (RSOS-181116.R1)

03-Jan-2019

Dear Dr Kang:

Title: Eco-friendly synthesis of VO₂ with stripped pentavalent vanadium solution extracted from vanadium-bearing shale by hydrothermal in high yield
Manuscript ID: RSOS-181116.R1

It is a pleasure to accept your manuscript in its current form for publication in Royal Society Open Science. The chemistry content of Royal Society Open Science is published in collaboration with the Royal Society of Chemistry.

On behalf of the Subject Editor Professor Anthony Stace and the Associate Editor Professor Eva Hevia.

RSC Associate Editor:
Comments to the Author:
(There are no comments.)

RSC Subject Editor:
Comments to the Author:
(There are no comments.)

Reviewer(s)' Comments to Author:
Reviewer: 1

Comments to the Author(s)
Authors have addressed all the comments and now paper is suitable to accept as is.

Reviewer: 2

Comments to the Author(s)
Authors have answered all my question in the peer review time. Hence, I recommend it for publication.

Appendix A

Comments on # RSOS-181116

Authors synthesized the VO₂(B) nanorods by an efficient and eco-friendly hydrothermal process with stripped vanadium solution. Various characterization techniques were used in order to find the structure, morphology of the sample such as XRD, Raman spectrum, SEM to show the as-synthesized VO₂ is a pure phase. Further materials were tested for cathode electrode in Li-ion batteries, which exhibit an initial specific capacity of 192.3 mAh/g and its capacity decreased to 95 mAh/g after 20 cycles.

[1] Authors should add clear TEM, HRTEM, SAED analysis of these nanorod samples.

[2] What is the long term cyclic stability of the materials ?

[3] Detail analysis of XPS needed in order to support the supercapacitor performance results.

[4] Have authors fabricated coin cell device or actual practical device?

Authors can add this data.

[5] Some of the relevant literature data need to be added. For example these important references are missing in the manuscript. *Journal of Alloys and Compounds* 695, (2017) 154-161; *Microporous and Mesoporous Materials* 244, (2017) 101-108; *Applied Surface Science* 418, (2017) 2-8; *RSC Advances* 5 (2015), 80990-80997; *European Journal of Inorganic Chemistry* 2015 (11), 1973–1980. *RSC Advances* 5, (2015) 88796 – 88804; *Journal of Physics and Chemistry of Solids* (2018), DOI: 10.1016/j.jpccs.2018.02.020

Appendix B

Point-by-point responses to the referees' comments

Dear Editor and Reviewers,

Thank you for your letter and for the reviewers' comments concerning our manuscript. The authors thank cordially the editor and two anonymous reviewers for giving us an opportunity to improve the manuscript. These comments are all valuable and very helpful for revising and improving our paper, as well as the important guiding significance to our research. We studied your comments carefully and have made revisions thoroughly which we hope meet with your approval. The specific corrections in the manuscript and the responds to the reviewer's comments are as follows:

Responses to Reviewer's comments:

Reviewer #1: Authors synthesized the VO₂(B) nanorods by an efficient and eco-friendly hydrothermal process with stripped vanadium solution. Various characterization technique were used in order to find the structure, morphology of the sample such as XRD, Raman spectrum, SEM to show the as-synthesized VO₂ is a pure phase. Further materials were tested for cathode electrode in Li-ion batteries, which exhibit an initial specific capacity of 192.3 mAh/g and its capacity decreased to 95 mAh/g after 20 cycles.

1. Authors should add clear TEM, HRTEM, SAED analysis of these nanorod samples.

Response: Thank you very much for your suggestions. According to your advice, TEM, HRTEM, SAED have been applied to analyze the microscopic morphology, electron diffraction, orientation of the crystal plane and the direction of growth of the material (shown in Fig. 7).

2. What is the long term cyclic stability of the materials ?

Response: Thanks for your suggestion. We have added the long term cyclic stability test of the materials and analyzed its performance (shown in Fig. 8a) in the revised manuscript.

3. Detail analysis of XPS needed in order to support the supercapacitor performance results.

Response: We agree with you that XPS could support the supercapacitor performance results. XPS test of the material and detail analysis have been added (Fig. 5 in the revised manuscript) in the revised manuscript.

4. Have authors fabricated coin cell device or actual practical device? Authors can add this data.

Response: Thanks for your constructive suggestion. I'm regret that we have not fabricated coin cell device or actual practical device. Our research group just started the study on the synthesis and electrochemical research of VO₂, and this research will be carried out in the near future.

5. Some of the relevant literature data need to be added. For example, these important references are missing in the manuscript. Journal of Alloys and Compounds 695, (2017) 154-161; Microporous and Mesoporous Materials 244, (2017) 101-108; Applied Surface Science 418, (2017) 2-8; RSC Advances 5 (2015), 80990-80997; European Journal of Inorganic Chemistry 2015 (11), 1973–1980. RSC Advances 5, (2015) 88796 – 88804; Journal of Physics and Chemistry of Solids (2018), DOI: 10.1016/j.jpcs.2018.02.020

Response: Thanks for your recommendation to cite these papers describing the electrochemical research. We read these papers carefully and have cited them in the revised manuscript.

Reviewer #2: After careful reading this manuscript, I cannot recommend this to be published in Royal society open science at the current form. However, it can be published after major revision if the authors addressed all the issues raised.

1. The experimental methodologies and material preparation are not new. Hence, a clear statement to the novelty of the work presented is necessary. Further, how did you choose 1073K for 1 h (stripped vanadium) and 220 °C for 8h (VO₂)? Are it's optimized conditions?

Response: Thank you very much for your careful review. Hydrothermal synthesis is commonly used to synthesize VO₂, but it is indeed that this method is the first time to be used to prepare VO₂ (B) with stripped pentavalent vanadium solution. Besides, stripped pentavalent vanadium solution is different from pure V₂O₅ as vanadium source because the former contains certain amounts of impurities such as Fe, Al, P, Si. The synthesized VO₂(B) obtained in this experiment is pure phase material, indicating that stripped pentavalent solution with a certain amounts of impurities also can be used to synthesize VO₂, which has never been reported before.

The vanadium-bearing shale for the preparation of stripped pentavalent vanadium solution is calcined at a temperature of 800 °C for 1 hour. Under this condition, vanadium can be effectively leached from the ore, which have been intensively studied by our group. The synthesis condition of 220 °C and 8 hours is obtained after the temperature and time exploration tests, which have been sorted and added in the revised manuscript.

2. The quality of fig. 3 (SEM pictures) is not sufficient for the publication.

Response: Thank you very much for your careful review. We have replaced Fig. 6 by a new one which is sufficient for the publication in the revised manuscript.

3. The electrode materials are characterized using a half-cell test configuration which is quite routine. Hence, authors need to provide the full device characterization for making the manuscript publishable.

Response: In the early literature review stage, we found that the half-cell is used to characterize the electrode material, and the full battery is used to study the overall system properties of the battery. What we want to investigate is the nature of the prepared VO₂ (B) as an electrode material, so the authors chose half-cell characterization.

4. The author had prepared their electrode material using 20 % acetylene black which will have a massive impact on the electrochemical measurement, So, we can't able to justify the true nature of the VO₂(B) as the cathode material.

Response: Because of acetylene black has the advantages of higher electronic conductivity, small particle size, and isotropy, the migration rate of electrons in the battery can be increased. Therefore, acetylene black is often used as a conductive agent for electrode materials. The choice of adding 20% acetylene black in this experiment is the result of reference to a large amount of relevant literatures.

5. The author needs to elaborate on the electrochemical method in the revised manuscript.

Response: Thanks for your suggestion. We have elaborated on the electrochemical method in the revised manuscript.

6. The mass loading of the electrode needs to be provided in the revised manuscript.

Response: Thank you very much for your careful review. The mass loading of the electrode has been provided in the revised manuscript.

7. Surface area and pore size analysis plays a vital role in the electrochemical properties of any material. Authors need to provide the BET analysis of the VO₂ in the revised manuscript.

Response: Thanks for your suggestion. We added BET test of the material and analyzed effect of surface area in the revised manuscript.

8. Long-term cyclic stability is one of the main aspects of electrode material to implement in the practical applications.so author needs to perform cyclic stability in the revised manuscript.

Response: Thanks for your suggestion. We added the long term cyclic stability test of the material and analyzed its performance (shown in Fig. 8a) in the revised manuscript.

9. The charge transfer kinetics and to evaluate the capacitive nature of the VO₂ electrode material EIS analysis need to be performed in the revised manuscript.

Response: Thanks for your suggestion. We added the EIS analysis (shown in Fig. 8e) of the material to describe the charge transfer kinetics and to evaluate the capacitive nature of the VO₂ electrode material in the revised manuscript.

10. Some literature about novel energy storage devices should be cited in suitable places.

References:

- 1) Journal of Industrial and Engineering Chemistry 64, (2018), 134-142.
- 2) Current Applied Physics 9(6), (2009), 1195-1198.
- 3) ChemElectroChem 4(12), (2017),3302-3308.
- 4) Electrochimica Acta 56(5), (2011), 2122-2126.
- 5) Cryst. Res. Technol. 46, (2011), 507 – 510.

Response: Thanks for your recommendation to cite these papers describing the electrochemical research. We read these papers carefully and have cited them in suitable places in the revised manuscript.

The authors thank again the editor and the reviewers for your constructive and pertinent comments, which have improved the manuscript to a large extent.

Corresponding author: Dr. Yimin Zhang. (zym126135@126.com)